# Shedding Light on Vitamin D Status and Its Complexities during Pregnancy, Infancy and Childhood: An Australian Perspective

**DOI:** 10.3390/ijerph16040538

**Published:** 2019-02-13

**Authors:** Nelfio Di Marco, Jonathan Kaufman, Christine P. Rodda

**Affiliations:** 1Women’s and Children’s Division, Sunshine Hospital, St Albans, VIC 3021, Australia; nelfiodm@gmail.com (N.D.M.); Jonathan.Kaufman@wh.org.au (J.K.); 2Department of Paediatrics, University of Melbourne, Royal Children’s Hospital, Parkville, VIC 3052, Australia; 3Australian Institute for Musculoskeletal Science, University of Melbourne, Sunshine Hospital, St Albans, VIC 3021, Australia

**Keywords:** Vitamin D deficiency, rickets, vitamin D during pregnancy, foetal life, infancy and lactation, UV B, sun exposure, health literacy, mineral ion nutrition, vitamin D dependent rickets

## Abstract

Ensuring that the entire Australian population is Vitamin D sufficient is challenging, given the wide range of latitudes spanned by the country, its multicultural population and highly urbanised lifestyle of the majority of its population. Specific issues related to the unique aspects of vitamin D metabolism during pregnancy and infancy further complicate how best to develop a universally safe and effective public health policy to ensure vitamin D adequacy for all. Furthermore, as Australia is considered a “sunny country”, it does not yet have a national vitamin D food supplementation policy. Rickets remains very uncommon in Australian infants and children, however it has been recognised for decades that infants of newly arrived immigrants remain particularly at risk. Yet vitamin D deficiency rickets is entirely preventable, with the caveat that when rickets occurs in the absence of preexisting risk factors and/or is poorly responsive to adequate treatment, consideration needs to be given to genetic forms of rickets.

## 1. Introduction

In the 1950s and 1960s, during the post-World War Two economic boom in Australia, rickets was considered a disease of the past in Australia, and in the global setting, a disease of poverty. During this era, the typical Australian lifestyle was characterised by active outdoor living, with many mothers able to stay at home with their children. A seemingly trivial lifestyle practice was that most household laundry was hung out to dry outside on most days on the back yard rotary clothes hoist, an unrecognised opportunity for regular sun exposure, before the advent of widespread use of clothes driers.

### 1.1. Traditional Practices for Prevention and Treatment of Rickets

Rickets was essentially first described contemporaneously by Daniel Whistler in a dissertation from the University of Leiden in 1645 and five years later by Cambridge Physician Dr Francis Glisson [1], who systematically described and published his personal observations [2,3]. Based on his observations from clinical examination and post-mortem findings, Glisson concluded that this condition was neither congenital nor contagious, but was essentially due to environmental factors. Furthermore, he had also accurately noted that rickets was rarely observed before six months of age and was predominantly a disease of infants and toddlers [3]. However, it took nearly another 300 years to elucidate the scientific underpinning of Glisson’s observations.

#### 1.1.1. Cod Liver Oil

The medicinal use of cod liver oil dates back to the 1700s [4]. Among people living in coastal areas, there was a long-standing appreciation in folklore of the medicinal benefit of cod liver oil, but the earliest recorded medicinal use of cod liver oil dates to 1789, is credited to Dr Darbey of the Manchester Infirmary [4], for his treatment of rheumatism. The recognition of cod liver oil as a specific remedy against rickets was noted as early as 1824 in the German medical literature. In 1861, Trousseau of France opined that rickets was caused by lack of sun exposure and a faulty diet, and that cod liver oil could effectively cure it [4]. Although it was clear to the medical and scientific community in the late 1800s that cod liver oil and sunlight exposure could cure rickets, it took until the early 1900s for researchers to discover that Vitamin D deficiency was the underlying cause of so called “nutritional” rickets.

Although similar in fatty acid composition to other fish oils, cod liver oil has higher concentrations of vitamins A and D. According to the United States Department of Agriculture, a tablespoon (13.6 g or 14.8 mL) of cod liver oil contains 4080 μg of retinol (vitamin A) [5] and 34 μg (1360 iu) of vitamin D [6]. The Dietary Reference Intake of vitamin A is 900 μg per day for adult men and 700 μg per day for adult women, while that for vitamin D is 5 µg (200 iu)–15 μg (600 iu) per day (doses increase with advancing years). The tolerable adult Upper Intake levels (ULs) are 3000 μg/day (vitamin A) and 100 μg (4000 iu)/day Vitamin D. The recommended daily intake of Vitamin A (as retinol equivalent) is 300–400 µg for infants and toddlers, 600–900 µg for male adolescents and 600–700 µ for female adolescents [5,6].

Whilst acknowledging that cod liver oil appropriately administered successfully prevented vitamin D deficiency rickets in the past, there is also a substantial risk of vitamin A intoxication without careful administration and its use to prevent vitamin D deficiency rickets can no longer be recommended.

#### 1.1.2. “Sunning”

Heliotherapy, or sunlight therapy has been used for centuries therapeutically and dates back to ancient Roman and Greek times [7]. In the first half of the 19th century sunlight was believed to have a role in the treatment of jaundice as well as in the treatment of rickets [7]. Placing a child in a room with sunlight exposure, for ten minutes at a time was commonplace [7,8], however this practice would have been ineffective in the prevention of rickets, as UVB is not transmitted through glass [9]. It has also been established that “sunning” babies for the purposes of jaundice treatment is not appropriate and is potentially harmful. Phototherapy administered under medical monitoring is required to treat neonatal jaundice.

In 1890, addressing the aetiology of rickets, Palm studied the relationship between the incidence of rickets and its geographical distribution, and concluded that rickets was caused by lack of exposure to sunlight. Palm also observed that despite a superior diet and relatively better sanitary conditions, infants residing in Britain were more at risk for rickets than those living in the tropics [2,3]. Exposure to plenty of sunshine, which was the norm for infants residing in the tropics, he proposed, was responsible for their protection against rickets. Palm recommended the “systematic use of sun-baths as a preventive and therapeutic measure in rickets” [3].

As skin cancer rates climbed steeply from the 1940s, the scientific and medical communities began to understand the potentially damaging effects of sun exposure to skin [2]. Although research has shown overwhelmingly that sunlight exposure is linked to skin cancer, ongoing “sunning” practices are still practised throughout the world [7,8,10]. Aladag and colleagues [7] found that although families and parents were aware of the benefits of sunlight exposure, there was a poor understanding of potential dangers and concerns around sunlight exposure. In general families were found to be “sunning” their babies for bone health, jaundice, nappy rash and to increase vitamin D. In geographical locations where health literacy and access to health care is low, “sunning” practices continue, likely to be related to the passing down of these practices throughout the generations.

## 2. A Resurgence of Rickets in Australia

In 1972 Mayne and McCredie reported a resurgence of rickets in Melbourne, recognising newly arrived southern European migrants and premature infants as the predominant risk group [11]. These authors reviewed radiological cases of rickets from 1961–1971 at the Melbourne Royal Children’s Hospital, and identified 59 cases of vitamin D deficiency rickets, and these comprised just over half the cases of radiological rickets identified. In the 1960′s vitamin D assays were not available [12] in public hospital settings in Melbourne, and the diagnosis was a clinical one, based on radiological findings without features of other underlying chronic conditions associated with rickets.

The Victorian State Government had established an infant welfare program for all Victorian infants and toddlers, under the directorship of Dr Vera Scantlebury-Brown in 1926. Under her leadership infant morbidity and mortality decreased dramatically [13]. Anecdotal evidence suggests that the use of cod liver oil to prevent rickets was also recommended around this time. However, most of the infants and toddlers described by Mayne and McCredie came from poor migrant families who did not attend maternal child health programs and furthermore had limited sun exposure. Although the need for vitamin D supplementation in premature infants is now well recognised [14,15], other environmental and lifestyle issues leading to vitamin D deficiency rickets were not considered a public health concern at the time. The authors’ abstract concluded with “A plea is made for adequate instruction of our migrant population in the prevention of this disease” [11].

Over the past 30 years there have been dramatic demographic and lifestyle changes which have had a major impact on the vitamin D status of the Australian population. Thankfully the White Australia Policy, which prevented migration of darker skinned people throughout most of the 20th century, was finally abolished in 1973. Yet once again over 20 years later in the 1990s, reports began to be published of rickets in Australian infants and children, predominantly from Melbourne [16,17,18]. At that time the only source of high dose vitamin D syrup to treat rickets was from the Royal Children’s Hospital pharmacy in Melbourne, suggestive of the perception that rickets was a rare disease in Australia. Most of the cases identified during this period were infants who were exclusively breastfed, and whose mothers had highly pigmented skin and/or were veiled, and were also vitamin D deficient [17,18], consistent with findings of Grover and Morley who reported that 80% of veiled or darker skinned pregnant women attending a tertiary Melbourne public hospital and tested, had a vitamin D of less than 22.5 nmol/L [19]. In 2012, Munns and colleagues [18] published a national audit of Vitamin D deficiency rickets in Australia identifying 398 vitamin D deficient (25 OH Vitamin D concentrations less than 50 nmol/L) infants and children aged less than 15 years. Most cases (251) were identified from Victoria, 69 from Western Australia and 70 from New South Wales. This equated to an annual incidence nationally of 4.9 cases/100,000. Eighty per cent of cases were identified through refugee screening programs and 81% of parents of the cases were highly pigmented from African nations [18].

More recently, two highly successful and important Australian public health campaigns were appropriately implemented, one was the “Breast is Best” campaign to encourage mothers to return to breast feeding, rather than providing infant formula feeding, and the other was the “Sun Smart” campaign, to protect the population from the harmful effects of excessive sun exposure [20]. However, both these campaigns have major implications for the maintenance of vitamin D adequacy.

Australia is considered a “sunny country”, so does not currently have national Vitamin D food fortification. Furthermore, developing a national vitamin D public health policy for pregnancy, infants and children in Australia, one of the most highly urbanised and multicultural continents globally, is complex. Since the abolition of the White Australia Policy in 1973, there is now a broad spectrum of skin colouring from very fair to very dark within the population [20]. Australia is also a large continent with latitudes ranging from 11° S in northern Australia to 43° S in Tasmania. Increasing urbanisation has resulted in some families spending less time participating in outdoor activities, in favour of indoor and screen-based activities. This lifestyle is also likely to be contributing to the increasing incidence of obesity (see “fat cells” upper right hand corner of Figure 1) [9], which is yet another risk factor for vitamin D deficiency, across all age groups in Australia. Consequently balancing Australia’s public health “Sun Smart” and “Breast is Best” messages with the prevention of vitamin D deficiency rickets is irrefutably challenging as discussed below.

## 3. Overview of Vitamin D Metabolism during Pregnancy, Infancy and Childhood

Human vitamin D metabolism involves multiple organ systems (Figure 1) [9], starting with conversion of 7-dehydrocholesterol (7DHC) to vitamin D within the epidermal layer of the skin. This reaction is catalysed by ultraviolet B with a wavelength of 290–315 nm [9,21,22]. Vitamin D then enters the circulation where it is rapidly 25-hydroxylated in the liver, where it remains as the storage form of vitamin D for several months. 25-hydroxy vitamin D (25 OH Vit D; calcidiol) is further hydroxylated to the “active” form of vitamin D, 1,25 dihydroxyvitamin D (1,25 (OH)_2_ Vit D; calcitriol), which should be considered a hormone rather than a vitamin, as it is tightly regulated within the kidney, by actions of PTH and FGF 23 on 1 alpha hydroxylase [22]. 1,25 (OH)_2_ Vit D acts on the gut to increase dietary calcium and phosphate absorption. Both 25OHVitD and 1,25 (OH)_2_Vit D are then inactivated by 24 hydroxylase to form 24,25 dihydroxyvitamin D 24,25 (OH)_2_ Vitamin D and 1,24,25 trihydroxyvitamin D (1,24,25 (OH)_3_ Vitamin D), respectively (Figure 1) [9].

In the modern developed world diet, few unsupplemented food groups contain sufficient vitamin D to provide a sole source of vitamin D (Table 1) [23], consequently approximately 90% of vitamin D is obtained from sunlight exposure. However, direct sunlight exposure is required, as UVB at this wavelength is not transmitted through glass or clothing [9]. Vitamin D synthesis in the skin is inversely proportional to melanin production, and higher skin melanin production results in darker skin, which provides protection from both sunburn and later development of skin cancer [2,24,25]. This skin protection however is counterbalanced by the need for individuals with highly pigmented skin to have up to approximately 6 times the sunlight exposure compared with fair skinned individuals to maintain a healthy vitamin D status [26,27]. UV B exposure also varies according to latitude, season and time of day. Consequently at higher latitudes UV B exposure is significantly greater during summer than winter (Figure 2), due to the relatively longer (oblique) passage of UV B through the earth’s atmosphere during winter compared with spring, so that at higher latitudes, most UV B is absorbed by the earth’s atmosphere during the winter months [9]. As a result, the greatest risk for vitamin D deficiency developing in such regions is late winter and early spring. Time of day is also important as solar radiation travels through the earth’s atmosphere at an oblique angle during the early morning and late afternoon, and as for seasonal variation of atmospheric UV B (Figure 2), very little or no UV B is transmitted through the earth’s atmosphere at the extreme ends of daylight.

Neonatal 25 OH Vitamin D concentrations reflect maternal stores during the second and third trimesters, as in utero the foetus receives 25 OH Vitamin D from transplacental transfer from the maternal circulation. As in postnatal life foetal 25 OH Vitamin D is stored in the foetal liver, and with maternal vitamin D adequacy, these stores typically sustain infant vitamin D adequacy until 3–4 months of age [28]. Breast milk however is a relatively poor source of vitamin D [29]. Issues related to maintaining vitamin D adequacy throughout the first year of life with exclusive breast feeding is discussed in more detail below in Section 5. Overall, infants most at risk of developing rickets are unsupplemented premature infants, and term infants living in industrialised cities with limited sun exposure, at high latitudes born to mothers at risk of vitamin D deficiency due to poor sunlight exposure, unfortified food sources, social isolation, increased skin pigmentation or cultural clothing practices, independently or in combination.

In addition to environmental factors influencing vitamin D metabolism, there are a number of congenital forms of rickets, which may develop due to mutations arising within the various hydroxylase genes involved with Vitamin D synthesis and catabolism, or involve the 1,25 (OH)_2_ Vitamin D receptor gene. Doubtless there are also numerous polymorphisms of these genes which may modify vitamin D requirements at an individual level. Furthermore, systemic disease, especially those affecting the liver, gut and kidney, and obesity are additional risk factors for vitamin D deficiency.

In summary Vitamin D metabolism is a complex interaction between both environmental and genetic factors, which are particularly important during pregnancy, infancy and growth in childhood. Given the multiple environmental, genetic and systemic influences on vitamin D metabolism, it is clear from a public health perspective, there is no easy “one size fits all” solution to rickets prevention, however through understanding vitamin D metabolism during pregnancy, infancy and childhood, and by recognising both individuals and defined populations at risk for developing rickets, rickets should once again become a disease of the past.

## 4. Vitamin D Assays

When reviewing vitamin D related clinical publications, it is critical to be able to appreciate the technical limitations and quality of 25 OH Vit D assays used at the time of publication. As reviewed in detail by Fraser and Milan in 2013 [12], measurement of 25 OH Vitamin D is technically difficult for many reasons. Firstly there are two forms of 25 OH Vit D: 25 OH Vitamin D_2_ (ergocalciferol) derived from dietary sources and 25 OH Vit D_3_ (cholecalciferol) which is endogenously produced from Vitamin D_3_ produced in the skin following UV B sun exposure. There are also technical difficulties arising from 25 OH Vit D being hydrophobic, unstable in water and strongly bound to the Vitamin D binding protein (VDBP) and direct natural light degrades 25 OH Vitamin D.

The earliest vitamin D assays developed in the 1970′s were Competitive Binding Protein (CBP) Assays, however they proved to be far too inaccurate when attempts were made to develop automated systems required for clinical use. Subsequently 25 OH Vitamin D radioimmunoassays were first developed in the 1980s, however there were also major limitations with radioimmunoassays, these included (1) measurement of both 25 OH Vitamin D_2_ and D_3_ with some assays predominantly measuring one or the other form, (2) various methods used to remove VDBP, (3) the high cross-reactivity with 24,25 (OH)_2_ vitamin D (inactive form) which may increase the reported 25 OHVitD by as much as 10–15 mmol/L and (4) presence of heterophile antibodies, which falsely elevate the amount of 25 OH Vitamin D detected [12].

Although the earliest liquid chromatography tandem mass spectroscopy (LC-MSMS) systems for measuring vitamin D metabolites were published in 1977, there were initially a number of technical issues with this method; however, these have gradually been addressed over time. LC-MSMS is able to provide simultaneous measurement of both 25 OH D_2_ and D_3_, to provide a total 25 OH Vitamin D. However it is critical to separate out the 3 epi-25 OH Vit D (an inactive form of vitamin D) which is typically seen in higher concentrations in neonates and young children. LC-MSMS is now viewed as the “gold standard” measurement of 25 OH Vit D.

In recent years the availability of Standard Reference Material (SRM) and the development of the Vitamin D External Quality Assurance Scheme (DEQAS) have significantly improved interlaboratory imprecision (CV), which was 32% in 1996 and 15.3% in 2009 [12]. However, in the routine clinical setting, 25 OHVitD immunoassays are still predominantly used as they currently provide more rapid and less expensive testing compared with LC-MSMS [12].

## 5. Vitamin D Physiology during Pregnancy and in the Foetus and Infant

Kovacs has published two complementary, extensive, scholarly reviews describing the physiology of mineral ion and vitamin D metabolism during pregnancy, foetal life and infancy [28,29], and these are briefly summarised below.

### 5.1. Pregnancy

Changes in maternal calcium metabolism during pregnancy are characterised by doubling or tripling of serum free (biologically active) calcitriol from the early first trimester, gradually increasing throughout pregnancy into the third trimester. The effect of pregnancy on 25 OH Vit D is less clear, likely due to environmental issues and assay variability between studies. Despite numerous reports of maternal vitamin D supplementation during pregnancy, many of which have been reviewed by Kovacs [29], some obstetricians remain concerned about the safety of vitamin D supplementation during pregnancy. Arguably one of the more robust studies was published by Hollis and colleagues [30]. They provided three doses of Vitamin D_3_, (400 iu (10 µg), 2000 iu (50 µg) and 4000 iu (100 µg)) to each of three groups within a total of 350 pregnant women of mixed ethnicity, from 12 to 16 weeks gestation until term delivery. These authors demonstrated that all three doses were safe to administer during pregnancy [30]. In addition to contributing to optimising maternal bone health during pregnancy, provision of vitamin D supplements to vitamin D deficient pregnant mothers particularly during the third trimester, will also optimise foetal hepatic vitamin D stores.

### 5.2. Foetus

At birth neonatal vitamin D concentrations reflect maternal vitamin D status, however prenatally foetal skeletal mineralisation occurs essentially independently of vitamin D and largely occurs during the third trimester when most of the 30 gm of skeletal calcium is accrued. Placental calcium transfer maintains circulating foetal calcium concentrations, and in contrast to extrauterine life, the foetal gut and kidney play a negligible role in calcium homeostasis in the foetus.

Foetal serum calcium concentrations are maintained 0.3–0.5 mmol/L higher than maternal concentrations by the foetal/placental hormone PTHrP [28]. PTHrP shares high homology with PTH within the first 36 N-terminal amino acids, consequently, PTHrP can also act via the PTH/PTHrP receptor on bone. However it is in its intact form that PTHrP maintains the foetal–maternal placental calcium gradient, a process independent of vitamin D status [28,31]. As illustrated in the lower left hand corner of Figure 1 [9], there are a number of examples of autocrine functions of 1,25 (OH)_2_ Vitamin D depicted, and there are likely to be additional autocrine effects in the developing foetus, including development of the immune system, affecting postnatal life [32,33,34,35] however detailed consideration of the role of vitamin D in the developing foetus independent of calcium and bone metabolism is beyond the scope of this paper.

### 5.3. Newborn

Cord blood vitamin D concentrations reflect maternal vitamin D status, as placental transfer of maternal vitamin D throughout the latter weeks of pregnancy is stored in the foetal liver [28]. The foetal–maternal calcium gradient maintained by PTHrP abruptly ends on delivery, with severing of the umbilical cord. The decrease in neonatal serum calcium stimulates the newborn parathyroid glands to switch from PTHrP to PTH secretion.

Gut absorption of calcium in preterm infants is almost entirely passive and independent of 1,25 (OH)_2_ [28]. Premature infants born less than 36 weeks gestation are at risk of developing osteopaenia [28,36], the severity of which increases with increasing prematurity, compounded by coexisting phosphate and calcium deficiency, if infants are not also supplemented with phosphate, with or without calcium supplementation as clinically indicated, and with Vitamin D supplementation [14,15,36].

Expression of intestinal Vitamin D Receptor increases towards term, although lactose in milk continues to facilitate passive gut calcium absorption in term infants [28]. This is a likely mechanism to explain why rickets is rarely seen within the first month or so of life in vitamin D deficient infants. Although calcium bioavailability is greater in breast milk compared with formula feeds, breast milk, even of vitamin D replete mothers, is a relatively poor source of vitamin D [29]. However, due to infant hepatic vitamin D stores, most fully breast infants of vitamin D replete mothers are expected to remain vitamin D sufficient for the first 3–4 months of life.

Various investigators have explored maternal rather than infant vitamin D supplementation [37,38,39,40]. As 25 OH Vitamin D is tightly bound to the VDBP, this results in a long circulating half-life of 4–6 weeks, however, this also results in little maternal circulating 25 OH Vitamin D being secreted into breast milk. In contrast, VDBP has a poor affinity for vitamin D_3_ allowing Vitamin D_3_ to be readily secreted into breast milk [39]. Yet because Vitamin D_3_ has a short circulating half-life of just 12–24 h, it must be given in relatively high, daily doses to provide exclusively breast fed infants with sufficient 25 OH Vitamin D. Hollis and colleagues conducted a randomised trial comparing administration of 400 iu Vitamin D to infants versus up to 6400 iu Vitamin D daily to lactating mothers, and found that this approach was safe and that 25 OH Vitamin D concentrations achieved in the infant were directly proportional to the amount of maternal supplementation [39]. In contrast, the use of high dose monthly vitamin D for example [38,40] would be expected to be less effective dosing regimens compared with daily dosing, due to the rapid hydroxylation of calciferol to 25 OH Vitamin D in the maternal liver subsequently preventing excretion into the breast milk [39].

## 6. Public Health Aspects of Vitamin D Deficiency: Vitamin D Adequacy and “Sun Smart” Behaviours

Most of the lifetime sun exposure occurs in childhood and it is known that early life exposure to sunlight can increase skin cancer risk later in life. Furthermore, repeated episodes of sunburn at any age increases skin cancer risk [2,10]. It is for this reason that dermatologists have recommend that vitamin D supplementation and diet should be the focus of prevention and treatment for vitamin D deficiency, rather than advocating increasing sunlight exposure, although there are published recommended safe sun exposure times with respect to latitude and the season [9].

“Sun Smart Victoria” [41] recommends that sun exposure between May to mid-August is unprotected unless for prolonged periods or when in areas that are highly reflective like the snow in alpine regions during winter. In contrast, in mid-August until the end of April only a few minutes daily in the sun is recommended without protection, otherwise full sun protection is recommended for longer periods [20]. On the contrary, for those with much darker skin types it may not be possible to maintain vitamin D levels from May to mid-August with sun exposure alone and vitamin D supplementation may be required (Figure 3) [20]. There is also ongoing development of sunscreens which transmit the specific UVB rays for vitamin D synthesis, with minimal dermal erythema, as described by Kockott and colleagues [42].

Although the Sun Smart Australia “Slip (on a shirt), Slop (on sunscreen), Slap (on a hat)” public health campaign has been resoundingly effective, because of the multiple and varied risks involved with developing vitamin D deficiency, it remains challenging to develop an effective national vitamin D deficiency prevention policy in Australia in this context.

## 7. Clinical Features of Rickets

### 7.1. Normal Bone Physiology

Bone is a complex structure and in growing infants and children undergoes constant “modelling”. Infant long bones comprise the diaphysis, metaphysis and epiphysis. Linear bone growth occurs at the metaphyseal growth plate within which chondroblasts produce bone matrix proteins (osteoid) on which phosphate precipitates. Calcium then coprecipitates with phosphate to form hydroxyapatite crystals, which then form on the helical collagen structure (osteoid) [43,44]. Bone tensile strength is provided by the bone collagen matrix and bone rigidity by mineralisation, to provide the strong, rigid structure of bone, required for weight bearing. Calcium and phosphate adequacy are predominantly regulated by 1,25 (OH)_2_ vitamin D, thus optimum bone mineralisation in the growing skeleton requires both adequate dietary calcium and phosphate, and vitamin D sufficiency.

### 7.2. Rickets

The likelihood of vitamin D deficiency rickets developing in infants and toddlers is reliably assessed by identifying well recognised risk factors including maternal vitamin D deficiency during pregnancy (for infants aged less than four months), prolonged exclusive breast feeding, highly pigmented skin, and minimal direct sun exposure [45,46]. Routine supplementation to prevent the development of vitamin D deficiency and rickets in these infants and toddlers, without specific vitamin D testing, is recommended in Australia [45]. A more recent Global Consensus for the prevention of rickets has been published by Munns and colleagues in 2016 [47].

Rickets is characterised by defective mineralisation of the epiphyseal growth plates [43,47]. Growth of unmineralised osteoid at the growth plate results in soft, compressible bone, which leads to “flaring” of wrists and ankles on weight bearing when crawling or walking. Consequently, wrist and ankle flaring or swelling is typically not seen in non-weight bearing infants aged less than six months. Dietary calcium and phosphate adequacy is predominantly regulated by 1,25 (OH)_2_ vitamin D, thus bone mineralisation in the growing skeleton may be impaired by either vitamin D deficiency or dietary calcium, or due to a combination of these deficiencies [48,49], except in rapidly growing premature infants not provided with mineral ion supplementation, when phosphate demands exceed dietary phosphate supply [36]. However, phosphate depletion may also be related to FGF 23 related genetic defects, such as X-linked hypophosphataemic rickets (see Section 8).

The features of vitamin D deficiency rickets [46,47] are age-dependent. Young infants commonly present with hypocalcaemic seizures and characteristic examination findings include a widened anterior fontanelle, frontal bossing, craniotabes (soft, readily depressible skull bones) and rachitic rosary (swelling of the costochondral junctions, felt anteriorly in the mid-clavicular line). Toddlers may present with delayed walking, and vitamin D deficiency should be considered in any child with leg bowing not walking by 18 months. “Physiological bowing” warrants further investigations if the intercondylar distance exceeds 5 cm, with the medial malleoli in direct contact [46]. Delayed closure of the anterior fontanelle may also be present. Older children typically complain of muscle pain and weakness and usually also have leg bowing. Poor linear growth may be present at any age.

Radiological features characteristic of Vitamin D deficiency rickets include metaphyseal widening (flaring), with cupping and frayed metaphysis [46] (Figure 4). However, rickets secondary to phosphate depletion related to FGF 23 related genetic defects, such as X-linked hypophosphataemic rickets (see Section 8), may be suspected radiologically as the metaphysis may be widened and undermineralised, without the irregular appearance of “fraying” which occurs with secondary hyperparathyroidism, almost invariably present in vitamin D deficiency rickets, but typically absent in X-linked hypophosphataemic rickets for example. Rachitic rosary with widening and cupping of the costochondral junctions may be observed incidentally on a chest X-ray performed for other reasons. Rachitic changes may also be seen on long bone X-rays performed in the setting of a fracture [17].

## 8. Congenital (Genetic) Forms of Rickets

Genetic forms of rickets are relatively rare [50,51,52] but need to be considered if rickets is present without any apparent risk factors nor any or minimal response to usual vitamin D dosing. Hypophosphataemic forms of rickets are characterised radiologically by undermineralised splayed metaphyses without “fraying” [43], with normal serum calcium and PTH concentrations and a decreased Tubular Reabsorption of Phosphate (TRP).

Vitamin D related gene mutations are described as Vitamin D Dependent Rickets (VDDR) and VDDR Type 1 gene defects are related to defects in the synthesis of 1,25-dihydroxyvitamin D and VDDR Type 2 defects are due to defects in vitamin D receptor activation. Both VDDR Types 1 and 2 are autosomal recessively inherited. A recently reported upregulated *CYP3A4* mutation (previously unrecognised as having a role in Vitamin D catabolism) has been shown to increase hepatic catabolism of both 25 OH and 1,25 (OH)_2_ Vitamin D, and it is proposed that this form of congenital rickets due to increased Vitamin D catabolism be referred to as VDDR Type 3 [52], and this form is autosomal dominant.

Mutational analysis to define the underlying gene defect in Vitamin D metabolism is critical to inform appropriate treatment.

## 9. Maintaining Vitamin D Adequacy during Pregnancy and Breast Feeding

Paxton and colleagues published evidenced based Australian clinical guidelines for maintaining Vitamin D adequacy throughout pregnancy, infancy and childhood in 2013 [45] and these remain a safe and effective approach for the prevention of rickets, despite some variations between these guidelines and the more recently published Global Consensus by Munns and colleagues [47]. As circulating Vitamin D is also regulated by the inactivating enzyme 24 hydroxylase, typically wide variations in Vitamin D supplementation or sun exposure still results in similar circulating 25 OH Vitamin D concentrations. In summary, Paxton and colleagues adopted a relatively conservative approach recommending that exclusively breastfed infants should have sun protection and minimal sun exposure for the first 6–12 months and should receive 400 iu (10 µg) Vitamin D daily. Pregnant women are advised to have 400–600 iu (10–15 µg) Vitamin D. Toddlers and older children to 18 years of age who have minimal sun exposure are also recommended to have Vitamin D 400 iu (10 µg) daily [45].

Sunburn is always to be avoided, but outside active play and physical activity is to be encouraged for general health benefits, in addition to achieving Vitamin D adequacy. The amount of time spent without sunscreen and other sun protection varies according to season, latitude, skin colour and other factors as discussed above.

## 10. Treatment of Vitamin D Deficiency Rickets

The evidenced based treatment regimens outlined by Paxton and colleagues [45] have been shown to be safe and effective in prevention of rickets, either as daily dosing or as a single bolus dose over 3 months of age. Bolus dosing is not recommended under 3 months of age, because of the small risk of inadvertently revealing downregulating *CYP 24A1* (24 hydroxylase) gene mutations, resulting in hypercalcaemia [53,54]. It should also be noted that supplements containing vitamin D_3_ should be used as they are more effective than vitamin D2 containing preparations [55].

In adequately treated rickets biochemical and radiological features resolve within 3 months (Figure 4). However in toddlers’ residual bowing, with healing of rickets demonstrated radiologically, may take a year or more to correct, and typically parents need to be reassured that no orthopaedic intervention is required. If 25 OH Vitamin D concentrations have normalised and radiological rickets persist with a raised alkaline phosphatase, consider severe dietary calcium deficiency. A validated, rapid dietary calcium assessment for use in children has recently been published by Nordblad and colleagues [56].

## 11. Conclusions

Ensuring vitamin D adequacy within the whole Australian population nationally remains problematic, particularly during pregnancy, infancy and childhood, despite the increased knowledge of the unique aspects of vitamin D physiology during these critical life stages. There needs to be acknowledgement that a simple public health message such a “Breast is Best” will not suffice for ensuring Vitamin D adequacy during these life stages.

As a major at risk group, refugees/immigrants with poor literacy generally and poor health literacy specifically remain a challenging group to ensure their Vitamin D adequacy. Once again maternal health nurses and other health professionals involved with refugee/immigrant health need to be specifically trained to communicate Vitamin D health messages simply and effectively to these groups.

Recent studies have demonstrated an association between vitamin D deficiency and other conditions, independent from the direct role of vitamin D in calcium and bone metabolism. Vitamin D deficiency during pregnancy has been associated with increased risk of pre-eclampsia [57] and an increased risk of asthma [34] and type 1 diabetes [58] in the offspring of vitamin D deficient mothers, warranting further research to determine possible underlying mechanisms. Ensuring vitamin D adequacy through maternal oral supplementation of vitamin D_3_ at a level of 80 µg (3200 IU) per day is safe and would prevent most forms of rickets as well as reducing incidence of some complications of pregnancy that are harmful to the mother and foetus. This level of supplementation also has been reported to be associated with reduced risk of heart disease, type 1 diabetes, and several forms of cancer in adults [59].

In consideration of the multiple environmental and genetic factors which influence vitamin D metabolism, unlike the very successful Australian public health “Breast is Best” and “Sun Smart” Campaigns, the “Holy Grail” of a similar simple public health message to ensure Vitamin D adequacy is likely to remain elusive. Perhaps consideration could be given to developing a risk calculator for vitamin D deficiency during pregnancy, infancy and childhood such as has been developed for osteoporosis (the FRAX calculator [60] for example), for Type 2 Diabetes Risk [61] and for vitamin D deficiency risk in adults, Annweiler and colleagues have recently developed an adult self-assessment tool [62]. Returning to Maine and McCredie’s “A plea is made for adequate instruction of our migrant population in the prevention of this disease” [11], given Australia’s multicultural population and highly urbanised lifestyles, and that all mothers and their babies should have access to a Maternal Child Health Nurse, there is a pressing need to develop a Maternal Child Health Vitamin D deficiency risk assessment tool, which could also be universally applied.

## Figures and Tables

**Figure 1 ijerph-16-00538-f001:**
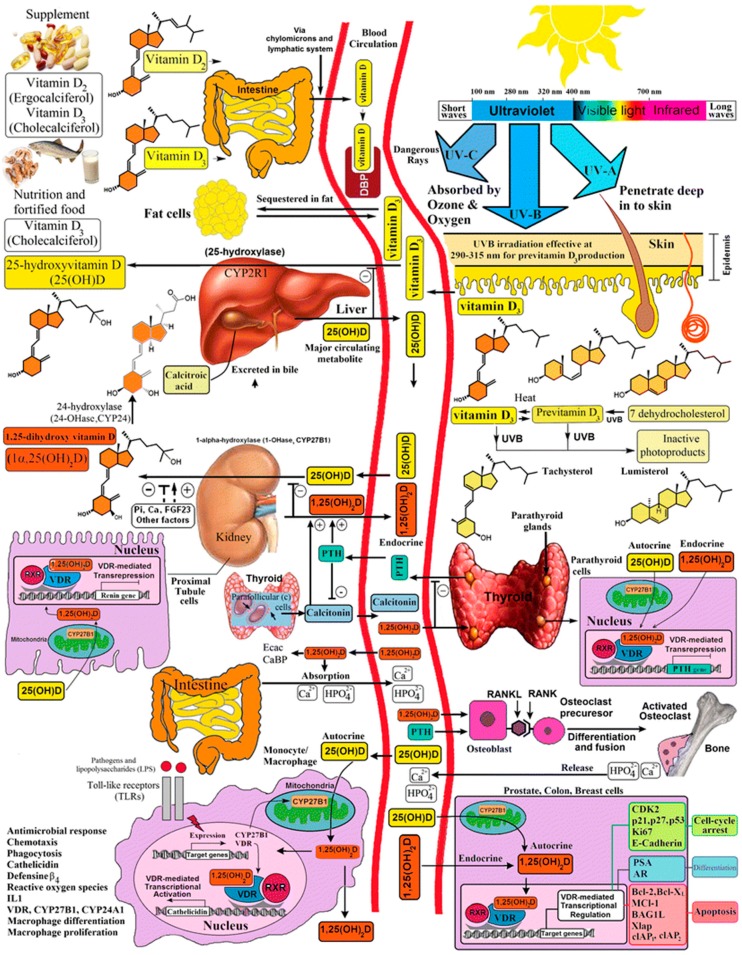
This schematic diagram highlights the complex interplay between a number of organs, and circulating forms of vitamin D, calcium, phosphate, and hormones involved with vitamin D metabolism and calcium homeostasis (shown within the two continuous red lines representing blood circulation). The top right hand corner depicts skin sun UV B exposure and vitamin D production in the skin, followed by 25 hydroxylation in the liver producing the storage form of vitamin D, 25 OH vitamin D, which is the form of vitamin D routinely measured in commercial assays. Inflammatory and paracrine roles of 1,25 (OH)_2_ vitamin D are also shown in the bottom left hand corner. Copyright permission was obtained from the publisher [9].

**Figure 2 ijerph-16-00538-f002:**
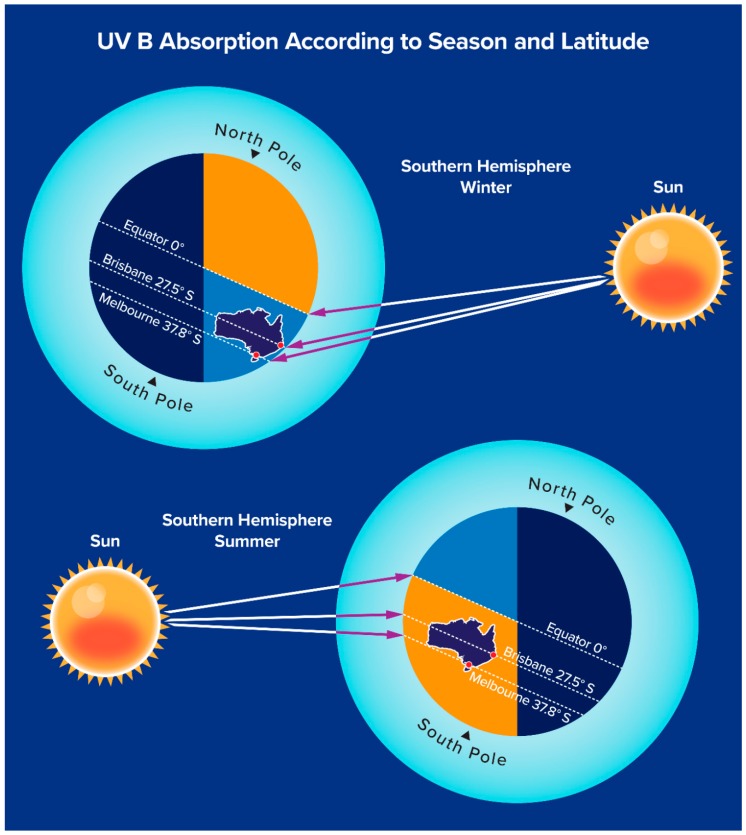
The earth’s atmosphere absorbs increasing amounts of UVB with increasing distance travelled. This schematic diagram (not drawn to scale) illustrates how the distance travelled through the earth’s atmosphere varies according to season and latitude, and that in winter this distance is relatively greater in Melbourne compared with Brisbane.

**Figure 3 ijerph-16-00538-f003:**
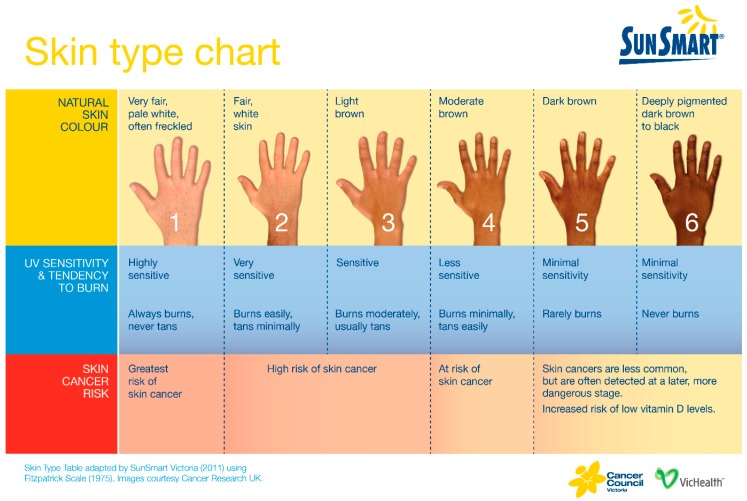
“Sun Smart” Guidelines for sun exposure according to skin type [20]. Copyright permission was obtained from the publisher.

**Figure 4 ijerph-16-00538-f004:**
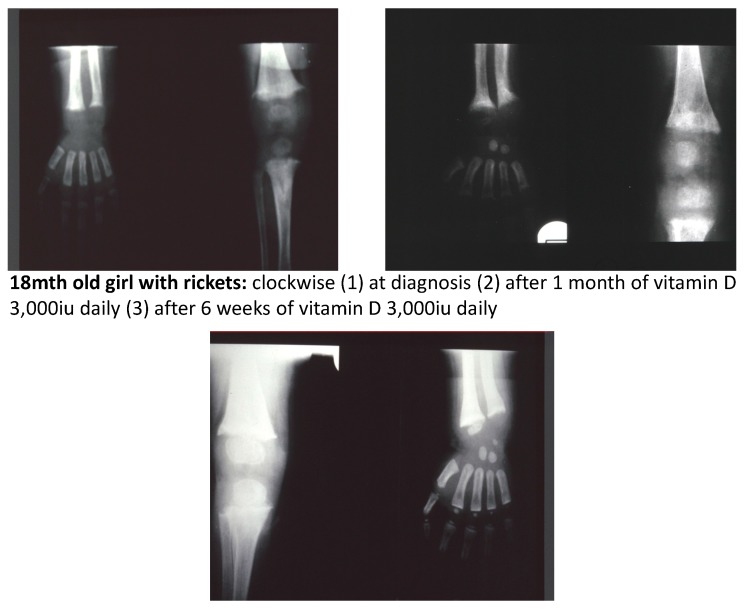
Radiographs of an 18-month-old girl demonstrating the relatively rapid metaphyseal mineralisation within 6 weeks in response to vitamin D treatment and adequate dietary calcium.

**Table 1 ijerph-16-00538-t001:** Dietary, supplemental and pharmaceutical sources of vitamin D_2_ and D_3_ adapted from Holick [23]. Copyright permission was obtained from the publisher.

Source	Vitamin D Content
***Natural sources***	
*Fish*	
Salmon (canned: 100 g)	Approximately 30–600 iu (7.5–15 µg) vit D_3_
Sardines (canned: 100 g)	Approximately 300 iu (7.5 µg) vit D_3_
Mackeral (canned: 100 g)	Approximately 250 iu (6.25 µg) vit D_3_
Tuna (canned: 100 g)	Approximately 230 iu (5.75 µg) vit D_3_

*Cod liver oil* (1 tsp/5 mL)	Approximately 400–1000 iu (10–25 µg) vit D_3_

*Shitake mushrooms*	
Fresh (100 g)	Approximately 100 iu (2.5 µg) vit D_2_
Sundried (100 g)	Approximately 1600 iu (40 µg) vit D_2_

*Egg yoke*	Approximately 20 iu (0.5 µg) vit D_2_ or D_3_

*UV B sun exposure* (0.5 minimal erythematous dose: equivalent to approximately 5–10 min or more of direct sun exposure to arms and legs depending on skin colour [Figure 3], latitude, season and time of day)	Approximately 3000 iu (75 µg) vit D_3_
***Infant feeding formulas***	400iu (10µg)/L vitamin D_3_
***Supplements***	
*For example:* Ostelin ©	1000iu (25µg) vitamin D_3_

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
