# Peer review of "Shedding Light on Vitamin D Status and Its Complexities during Pregnancy, Infancy and Childhood: An Australian Perspective"

_ijerph, 2019, doi:10.3390/ijerph16040538_

Reviewer 1 Report

The article deals with the contemporary Australian problem of prevention and therapy of rickets presented on the background of the historical development of knowledge about the causes of rickets formation and its treatment. The article is interesting, it makes aware  how much the problem of rickets prevention is geographically and ethnically conditioned and must be resolved respectively to local environment. 

I have some comments to  the text.

Tables and Boxes should not be pasted directly from other publications. This procedure makes that, among other imperfections,   tables have two titles.

I did not find in the text information justifying the inclusion of table 2.

The title of Box 1 does not correspond to its content, which is the same as in Box 3.

The Chapter 3 – “Vitamin D assays” should be after Chapter 4 – “Overview of vitamin D metabolism (…)”.

Author Response

30th  January 2019

Reviewer 1

Special Issue:  Vitamin D and Public Health

IJERPH

Dear Reviewer,

RE:  Manuscript submitted “Shedding Light on Vitamin D Status and its complexities during pregnancy, infancy and childhood:  an Australian perspective”;  Authors:  Nelfio Di Marco, Jonathan Kaufman and Christine P. Rodda* (*Corresponding author)

Thank you for your helpful comments.

As you can see from our revised manuscript, changes recommended by both reviewers have been incorporated and are highlighted in yellow.

Thank you for your comments relating to the illustrations.  On reflection we have decided to delete several of them as they do not add sufficiently to the text.  The labelling of the remaining illustrations is now correct.

We too had wondered about the ordering of sections 3 and 4 and this has now been reversed in the revised manuscript.

Yours sincerely,

A/Professor Christine Rodda

Reviewer 2 Report

This well-written manuscript provides background on the challenges of developing a public health policy for vitamin D supplementation in Australia.  The article acknowledges the gradient in latitude and UVB intensity across the country.  The authors also recognize the risks to vitamin D sufficiency posed by recommendations of sun avoidance and sunscreen use, with the goal of reducing risk of skin cancer.   The authors correctly warn against reliance on cod liver oil as supplement to treat vitamin D deficiency. 

The authors also should warn against supplements containing vitamin D2 (ergocalciferol) which are not as effective as vitamin D3 (cholecalciferol) in preventing vitamin D deficiency associated disease. The authors correctly point out the near impossibility of obtaining adequate vitamin D levels from food sources.  The paper also reports a recurrence of rickets in Australia in 1972  and successful efforts to control it.

General Recommendations:  While providing an interesting historical overview of rickets, this paper focuses narrowly on rickets as the primary complication of vitamin D deficiency.  In addition to rickets, there is compelling evidence that vitamin D deficiency also is associated with gestational diabetes, preeclampsia and other complications of pregnancy, as well as with asthma and type 1 diabetes in offspring.   In adults, serum levels of vitamin D less than 40 ng/ml are associated with increased risk of breast and colon cancer.  To address the public health issue of how much vitamin D is optimal, additional disease endpoints should be considered.   The amount of vitamin D required to prevent rickets for example will be less than the amount needed to optimize bone health of the mother or reduce risk of type1 diabetes or cancer. 

Specific Recommendations: It has been shown that mothers can safely be supplemented with up to 4,000 IU (100ug) of vitamin D3 per day and the authors cite a study supporting this (line 250, Hollis).  Currently, the  Australian Ministry of Health recognizes that daily intake of 80ug or 3,200 IU of vitamin D3 per day is safe for pregnant and lactating mothers as well as for children and adults, and that infants, 0-12 months, may safely be supplemented at 25 ug/day (1,000 IU).    

These figures are conservative but are a matter of policy, so the conclusion of this paper should state explicitly that:  "Ensuring vitamin D adequacy through maternal oral supplementation of vitamin D3 at a level of 80ug (3,200 IU) per day is safe and would prevent most forms of rickets as well as reducing incidence of some complications of pregnancy that are harmful to the mother and fetus.  This level of supplementation also has been reported to be associated with reduced risk of heart disease, type 1 diabetes, and several forms of cancer in adults."     

Some relevant references are provided below. 

Minor suggestion and historical note: It may be more appropriate to credit Daniel Whistler with providing the first description of rickets.  Francis Glisson published his treatise in 1650, five years after Whistler's dissertation from the University of Leyden was published in 1645 in which he described many of the clinical features of rickets ("De morbo puerili Anglorum")

Clarke, E.. "Whistler and Glisson on Rickets." Bulletin of the History of Medicine, vol. 36, no. 1, 1962, pp. 45-61. JSTOR, www.jstor.org/stable/44449766.

Daniel Whistler (1619-1684) The Rickets. JAMA. 1968;205(7):526. doi:10.1001/jama 1968.03140330068016

Line 40: Typo ...underpinning of Glisson's (not Glasson's) observations

Relevant Additional References: 

Vieth R. Why the optimal requirement for vitamin D3 is probably much higher than what is officially recommended for adults. J Steroid Biochem Mol Biol 2004;89-90:575-9

Heaney RP, Recker RR, Grote J, Horst RL, Armas LA. Vitamin D(3) is more potent than vitamin D(2) in humans. J Clin Endocrinol Metab. 2011;96(3):E447-52. doi: 10.1210/jc. 2010-2230. Epub 2010 Dec 22.

PMID: 21177785

Gernand AD, Simhan HN, Baca KM, Caritis S, Bodnar LM. Vitamin D, pre-eclampsia, and preterm birth among pregnancies at high risk for pre-eclampsia: an analysis of data from a low-dose aspirin trial. BJOG. 2017;124(12):1874-1882. doi: 10.1111/1471-0528.14372. Epub 2016 Oct 5.

Pilz S, Zittermann A, Obeid R, Hahn A, Pludowski P, Trummer C, Lerchbaum E, Pérez-López FR, Karras SN, März W. The Role of Vitamin D in Fertility and during Pregnancy and Lactation: A Review of Clinical Data. Int J Environ Res Public Health. 2018 Oct 12;15(10). pii: E2241. doi: 10.3390/ijerph15102241.

McDonnell SL, Baggerly C, French CB, Baggerly LL, Garland CF, Gorham ED, Lappe JM, Heaney RP.  Serum 25-Hydroxyvitamin D concentrations ≥40 ng/ml are associated with >65% lower cancer risk: pooled analysis of randomized trial and prospective cohort study. PLoS One. 2016 6;11(4):e0152441. doi: 10.1371/journal.pone.0152441.

Baggerly CA, Cuomo RE, French CB, Garland CF, Gorham ED, Grant WB, Heaney RP, Holick MF, Hollis BW, McDonnell SL, Pittaway M, Seaton P, Wagner CL, Wunsch A.  Sunlight and Vitamin D: Necessary for Public Health.  J Am Coll Nutr. 2015;34(4):359-65. doi: 10.1080/07315724.2015.1039866. Epub 2015 Jun 22.

Liu C, Wang J, Wan Y, Xia X, Pan J, Gu W, Li M. Serum vitamin D deficiency in children and adolescents is associated with type 1 diabetes mellitus.  Endocr Connect. 2018 Oct 1. pii: /journals/ec/aop/ec-18-0191.xml. doi: 10.1530/EC-18-0191. [Epub ahead of print]

Author Response

30th  January 2019

Reviewer 2

Special Issue:  Vitamin D and Public Health

IJERPH

Dear Reviewer,

RE:  Manuscript submitted “Shedding Light on Vitamin D Status and its complexities during pregnancy, infancy and childhood:  an Australian perspective”;  Authors:  Nelfio Di Marco, Jonathan Kaufman and Christine P. Rodda* (*Corresponding author)

Thank you for your most helpful and extensive comments, and generous provision of additional references.

As you can see from our revised manuscript, changes recommended by both reviewers have been incorporated and are highlighted in yellow.

Firstly, a reference to Daniel Whistler has now been included and the mis-spelling of “Glisson” has been corrected. 

More broadly, we do acknowledge that the main focus of this article is prevention of rickets in infants and children, and our reason for this is that we wanted to provide a background detailing the unique aspects of vitamin D metabolism during pregnancy, lactation, infancy and childhood, specifically including transplacental passage of vitamin D, breast milk as a poor source of vitamin D and the effects of vitamin D deficiency on the growth plate for example, rather than focussing more on associated long term health issues such as cancer and cardiovascular disease in older adults.  Although as you have indicated there are also wider health implications associated with in utero and postnatal vitamin D deficiency during childhood that have recently been described, although the underlying immune and other mechanisms are still largely not well understood, however, we agree that these issues do need to be briefly discussed.  As you can see from our revised manuscript, we have now included these comments.

Thank you for your suggestion of the following wording:  "Ensuring vitamin D adequacy through maternal oral supplementation of vitamin D3 at a level of 80ug (3,200 IU) per day is safe and would prevent most forms of rickets as well as reducing incidence of some complications of pregnancy that are harmful to the mother and foetus.  This level of supplementation also has been reported to be associated with reduced risk of heart disease, type 1 diabetes, and several forms of cancer in adults."    And this has now been incorporated into the manuscript with the relevant references shown.

Yours sincerely,

A/Professor Christine Rodda
